# Mind the Gap—Deciphering GPCR Pharmacology Using 3D Pharmacophores and Artificial Intelligence

**DOI:** 10.3390/ph15111304

**Published:** 2022-10-22

**Authors:** Theresa Noonan, Katrin Denzinger, Valerij Talagayev, Yu Chen, Kristina Puls, Clemens Alexander Wolf, Sijie Liu, Trung Ngoc Nguyen, Gerhard Wolber

**Affiliations:** Department of Pharmaceutical and Medicinal Chemistry, Institute of Pharmacy, Freie Universität Berlin, Königin-Luise-Straße 2-4, D-14195 Berlin, Germany

**Keywords:** ligand-based pharmacophores, structure-based pharmacophores, virtual screening, drug design, machine learning, molecular dynamics, de novo design, GPCR

## Abstract

G protein-coupled receptors (GPCRs) are amongst the most pharmaceutically relevant and well-studied protein targets, yet unanswered questions in the field leave significant gaps in our understanding of their nuanced structure and function. Three-dimensional pharmacophore models are powerful computational tools in in silico drug discovery, presenting myriad opportunities for the integration of GPCR structural biology and cheminformatics. This review highlights success stories in the application of 3D pharmacophore modeling to de novo drug design, the discovery of biased and allosteric ligands, scaffold hopping, QSAR analysis, hit-to-lead optimization, GPCR de-orphanization, mechanistic understanding of GPCR pharmacology and the elucidation of ligand–receptor interactions. Furthermore, advances in the incorporation of dynamics and machine learning are highlighted. The review will analyze challenges in the field of GPCR drug discovery, detailing how 3D pharmacophore modeling can be used to address them. Finally, we will present opportunities afforded by 3D pharmacophore modeling in the advancement of our understanding and targeting of GPCRs.

## 1. Introduction

G protein-coupled receptors (GPCRs) are a functionally and topologically diverse superfamily of heptahelical transmembrane receptors heavily involved in a variety of physiological cellular processes. Due to this family’s rich assortment of sub-branches, GPCR ligands can comprise almost any chemical entity: peptides (such as angiotensin, glucagon, and endothelin), amino acids and their derivatives such as biogenic amines (including glutamate, norepinephrine, and histamine) and eicosanoids (such as prostaglandins and leukotrienes), to name but a few. GPCRs are of great pharmaceutical relevance in that their ligands account for 35% of currently marketed drugs [1]. Drugs targeting GPCRs are implicated in myriad disorders, from cardiovascular [2] and metabolic diseases [3] to cancer [4].

Based on their phylogenetic relation, GPCRs are commonly divided into the six groups A–F, or, more recently, into the five glutamate-like, rhodopsin-like, adhesion, frizzled and secretin-like (GRAFS) families [5]. Well-studied GPCRs such as the β-adrenergic or dopamine receptors possess a multitude of known ligands. On the other hand, there are also several GPCRs that essentially lack their endogenous ligands and are therefore referred to as orphan GPCRs. Considerable effort is being made to bring about their deorphanization as these GPCRs might prove to be viable targets in combating emerging diseases.

GPCR activation involves the binding of extracellular ligands and subsequent conformational changes, which allosterically favors intracellular binding of so-called transducers such as G proteins or β-arrestins by the mechanism of allosteric coupling. Transducer binding in turn sets off a spatio-temporally regulated cascade of changes in second messenger concentrations, including cyclic adenosine monophosphate (cAMP), diacetylglycerole (DAG) or calcium ions. The cooperation of ligands, receptors and transducers results in the activation or modulation of multiple signaling pathways, orchestrated by GPCRs [6].

Certain ligands activate only a part of the available signaling pathways, preferentially activating one pathway over the other, a process which has been coined biased agonism [6]. The activation of a GPCR by a biased agonist can induce a so-called functional selective receptor response. Since the activation of signaling pathway subsets can result in enhanced therapeutic efficacy, this still somewhat recent discovery is one of the factors drenching the field of GPCRs with opportunities for therapeutic breakthroughs and places GPCRs at the cutting edge of contemporary medicinal research. There is a sustained need in pharmaceutical sciences for novel drug entities with improved binding attributes to take on challenges in contemporary medicine, such as polypharmacy and orphan diseases [7].

One powerful tool to address this unmet need for novel GPCR ligands is the use of 3D pharmacophores. The pharmacophore concept has been developed and matured over the 20th century [8]. The International Union of Pure and Applied Chemistry (IUPAC) defines a pharmacophore as “the ensemble of steric and electronic features that is necessary to ensure the optimal supramolecular interactions with a specific biological target structure and to trigger (or to block) its biological response” [9]. Three-dimensional pharmacophore models represent comprehensive, intuitive and easy-to-use scaffolds that unveil novel active chemical entities and structure–activity relationships (SAR). Three-dimensional pharmacophores are frequently used for virtual screening (VS) campaigns in the identification of novel ligands. Other applications for 3D pharmacophores include the mechanistic elucidation of proteins, modeling their functionality, or characterizing distinct protein conformations.

Three-dimensional pharmacophores have helped in understanding how a ligand can influence the conformation of a GPCR. Exemplary ligand–protein interactions of active and inactive GPCR conformations described via 3D pharmacophores are compared in Figure 1. The M2 muscarinic acetylcholine receptor serves as a good example to illustrate distinct interaction patterns for agonists and antagonists. The ligands in this case exhibit different interaction patterns with the same residues at the same binding site indicating distinct binding properties for agonists compared to antagonists and portraying GPCR shape flexibility. Other advanced techniques to combat challenges in the prediction of GPCR ligand performance include machine learning (ML) methods with usage of pharmacophore-based descriptors [10] or the composition of dynamic pharmacophores (also termed dynophores); the advantages afforded by the use of 3D pharmacophores become even more blatant when analyzed in combination with molecular dynamics (MD) simulations.

Three-dimensional pharmacophores are distinguished into ligand-based 3D pharmacophores and structure-based pharmacophores. The respective associated methods are classed as ligand-based drug design (LBDD) and structure-based drug design (SBDD). Ligand-based pharmacophores are generated by a set of known active ligands (such as small molecules or peptides). Ligand-based pharmacophores can be derived from the structure of a single ligand, or from a set of ligands, in which case the features that all the ligands have in common are synthesized into a so-called shared-feature pharmacophore. The consolidation of pharmacophore features stemming from multiple ligands is termed a merged feature pharmacophore. Structure-based 3D pharmacophores on the other hand are derived from apo structures of the target (macromolecules such as proteins or nucleic acids) or ligand–target complexes [15]. The main advantage of ligand-based 3D pharmacophores is the possibility of handling targets that proven to be evasive to date; they are powerful tools when target sites are unknown or unstudied. When an atomistic model of the target is available, structure-based 3D pharmacophores can be generated from apo binding sites or from structures of ligand–target complexes for more accurate drug design. Such models are usually derived from X-ray crystallography, nuclear magnetic resonance (NMR) spectroscopy, cryo-electron microscopy (cryo-EM), homology modeling or machine learning prediction algorithms such as AlphaFold [16].

GPCRs have been subject to experimental testing for at least 50 years [17] and a plethora of drugs targeting GPCRs were available long before the relatively recent start of the elucidation of their 3D structures. GPCR 3D structure determination had long been obstructed by their hydrophobicity and attachment to the membrane. These obstacles were overcome for the first time with the pioneering resolution of the 3D structure of inactive rhodopsin bound to 11-*cis*-retinal in 2000 [18] (Figure 2). This led to the release of the crystal structure of human β2 adrenoceptor in an inactive-like conformation in 2007 [19,20,21], the first 3D X-ray crystal structure of a protein of pharmaceutical relevance. The first active-like crystal structure of β2 adrenoceptor was released four years later [22] and paved the way for the resolution of many more GPCR crystal structures. It was in 2012 that the first case of ligands retrieved from structure-directed drug design based on a GPCR crystal structure was reported [23,24]. The long-awaited advent of cryo-EM-derived structures deposited in the PDB was kicked off with the resolution of the structure of the Calcitonin receptor in 2017 [25]. In 2018 the unliganded Frizzled 4 receptor structure was resolved, representing the first ever GPCR structure of medicinal relevance in its apo conformation [26].

Since these ground-breaking successes, a total of over 300 GPCR 3D structures representing over 60 targets has been made publicly available [27]. Each GPCR can assume a notoriously wide variety of conformations corresponding to different intracellular effects. Their conformational spectrum spans a wide range of conformations between active-like and inactive-like. All these conformations are necessary to account for the full flexibility of a specific GPCR, therefore the persistent lack of active-like 3D structures for orphan GPCRs [28] continues to hamper their mechanistic elucidation.

Our literature search revealed the most common applications of 3D pharmacophores in the field of GPCRs to be 3D pharmacophore-based VS (PBVS) (Section 2), the generation of quantitative structure–activity relationship (QSAR) models (Section 3), and hit-to-lead optimization and scaffold hopping (Section 4).

The successful application of 3D pharmacophore modeling to GPCRs started in the 1990s with studies such as those illuminating the binding of dopamine D_2_ and serotonin 5-HT_1A_ ligands to their respective receptors on a molecular level [29]. The boom in structural GPCR elucidation greatly boosted the opportunities for innovation and creativity in computational investigations into GPCRs, such as the use of 3D pharmacophores. This review aims to re-frame a well-established class of pharmacological targets through the unique lens of 3D pharmacophore modeling. We will summarize and analyze the existing wealth of studies in which 3D pharmacophore-guided computer-aided drug design (CADD) is applied to GPCRs.

Furthermore, we will highlight exciting advancements in the incorporation of 3D pharmacophore modeling into dynamic investigations, ML, and artificial intelligence (AI). Challenges in the field will be addressed and solutions suggested, and potential avenues for future research will be explored.

## 2. Virtual Screening

Three-dimensional pharmacophores can be used as filters in VS to determine new ligands for therapeutic targets by investigating a library of chemical compounds for their ability to exhibit a desired interaction pattern to the target [30,31,32,33,34,35,36,37,38,39,40,41,42,43,44,45,46,47,48,49,50,51,52,53,54]. Compounds capable of fulfilling the interaction features specified in the screening pharmacophore are termed ‘hits’. In most cases a small number of hits are chosen for subsequent experimental testing to confirm pharmacological effects. Thus, VS serves as a filter to enrich the hit rate for experimental testing and significantly reduces the costs compared to high-throughput screening (HTS) [55].

PBVS is often complemented by molecular docking, a method by which the binding modes of compounds within the target binding site are predicted [56]. Docking can either be used to identify hits via docking-based VS (DBVS), or as a subsequent filter for VS hits derived by another query [56,57]. In fingerprint-based VS, hits are defined by physicochemical properties shared between known actives and potential drug candidates [58,59]. This review focuses on PBVS as a valuable and promising tool to investigate new drug candidates.

In cases where active or inactive compounds for the binding site in question are already known, the screening pharmacophore can be validated by its ability to distinguish between true active and inactive decoy molecules. For more information on 3D pharmacophore validation the reader is referred to our previously published review [15] or that of Braga and Andrade [60].

PBVS allows efficient investigation of the pharmacological space of orphan receptors, for which no or little structural data are known [39,41,42,44,48]. This approach can also be used in scaffold hopping, whereby new chemical scaffolds distinct to the already known ligands of a protein target are determined, which may exhibit increased pharmacological potency [33,34,35,36,39,48]. The range of applications of PBVS extends to the detection of ligands capable of activating distinct signaling pathways, such as biased ligands [35,61,62,63] or of binding to distinct sites, such as allosteric modulators [39,64]. Furthermore, this approach enables the efficient exploration of the therapeutic potential of medicinal plant compounds, which are often difficult to isolate or synthesize in reasonable amounts for HTS [35]. In this section, we aim to characterize PBVS as a valuable and promising tool to identify new drug candidates for GPCRs, highlighting recent breakthroughs in GPCR research within this methodology.

### 2.1. Orthosteric Ligands

#### 2.1.1. Bombesin Receptor 1

Rasaeifar et al. [53] constructed a homology model of the human bombesin receptor (BB1) with the use of the rat neurotensin receptor NTS1 (PDB ID: 4GRV) [49] as a template. The homology model was refined through MD simulations with the antagonist PD176252 bound to the orthosteric binding site of the receptor, an approach which is discussed further in Section 6. Known BB1 antagonists were docked into the orthosteric site. The binding hypotheses were ranked according to scoring functions and the selected hypothesis with the highest score underwent energy minimization. The analysis of the ligand–receptor complex in conjunction with available SAR and mutagenesis studies allowed the authors to propose plausible binding hypotheses of the antagonists bound to the BB1 orthosteric binding site. The binding hypothesis of PD176252 was used to generate a 3D pharmacophore, which was used in VS to identify a set of small molecules with high affinity toward the BB1 receptor.

#### 2.1.2. Bradykinin Receptors

Lupala et al. used the X-ray crystal structures of the CXCR4 chemokine receptor and bovine rhodopsin receptor (PDB ID: 3ODU [65] and 1GZM [66], respectively) as templates in the construction of a homology model of the human Bradykinin B1 receptor (hBKBR1) [51], as discussed further in Section 7.1. The authors then performed molecular docking of known hBKB1R antagonists. They superimposed the docked ligand–receptor complexes and derived shared-feature pharmacophores of the ligands in complex with the receptor. Thus, this 3D pharmacophore model described the features associated with hBKB1R antagonism. This pharmacophore was used to screen for novel hBKB1R antagonists, four of which displayed antagonism at the hBKB1R. A similar workflow was implemented for the structure-based VS for antagonists of the human Bradykinin B2 receptor (B2R) [52].

#### 2.1.3. β3-Adrenergic Receptor

The β3-adrenergic receptors (β_3_-AR) are an appealing target for novel pharmacological therapies for metabolic, cardiovascular, urinary, and ocular diseases. Ujiantari et al. conducted a prospective ligand-based pharmacophore-based VS for the sake of identifying β3-AR agonists that do not elicit adverse effects [67]. In total, 93 unique compounds showing β_3_-AR agonism were retrieved from databases and further split into two subsets that were used for pharmacophore generation. The 3D pharmacophore models were generated based on the training set which contains 11 known selective β_3_-AR agonists and validated by a testing set that includes 72 active compounds, four inactive compounds and 6229 decoys. The pharmacophore model that matched all query features was used in VS against the Specs [68] and Drugbank databases [69]. After physicochemical property filtering and homology-modeled structure-based docking evaluation, 35 compounds were selected for an in vitro assay that measured cAMP levels at the β_3_-AR. Finally, four compounds were shown to display agonist activity at the human β_3_-AR.

#### 2.1.4. Cannabinoid Receptor 2

Hu and colleagues constructed homology models of the active and inactive cannabinoid receptor 2 (CBR2) [33]. The group performed molecular docking of a known agonist into the active-state model and a known inverse agonist into the inactive state. They created one ligand-based 3D pharmacophore each from an agonist and an inverse agonist for VS. The agonist and inverse agonist hits were filtered via molecular docking into the active or inactive model, respectively. Binding to CBR2 was confirmed via radioligand binding assays and agonist or inverse agonist activities were tested in a cAMP assay. Of the inactive hits, one acted as an inverse agonist. One other hit acted as neutral antagonist at concentrations below 10 μm and as an inverse agonist above 10 μm in concentration.

#### 2.1.5. G Protein-Coupled Bile Acid Receptor 1

The G protein-coupled bile acid receptor 1 (GPBAR1), also known as TGR5, is implicated in diseases including atherosclerosis, type 2 diabetes, and obesity [70]. Kirchweger et al. constructed two separate merged feature 3D pharmacophores out of the structures of known GPBAR1 ligands and used them in VS of databases comprising natural products and synthetic small molecules [47]. The resulting hits were filtered in silico via clustering by physicochemical properties and by performing a shape-based comparison of the hit compounds to the structure of one of the known GPBAR1 antagonists used in 3D pharmacophore model generation. The remaining hits were tested experimentally in a reporter gene-based assay for the determination of their GPBAR1 activation ability. Two natural compounds displayed activity comparable to that of the natural GPBAR1 ligand lithocholic acid and seven further compounds featuring novel natural as well as chemical scaffolds showed moderate receptor activation [47].

Zhao and coworkers used the chemical structures of known TGR5 agonists to create a ligand-based pharmacophore [43]. The identification of novel TGR5 agonists was performed with a multi-step workflow, consisting of ligand-based pharmacophore screening, molecular docking of the hits into the receptor (PDB ID: 7CFM [71]) and further filtering via visual inspection. This resulted in the selection of 20 compounds for in vitro biological evaluation by their ability to activate TGR5 as per the assessed intracellular levels of cAMP. Two of the tested compounds displayed TGR5 agonist activity. These are potential TGR5 agonist candidates and can additionally be used as starting structures in hit-to-lead optimization in developing novel TGR5 agonists.

#### 2.1.6. G Protein-Coupled Estrogen Receptor

O’Dea et al. also made use of hybrid VS in their study on the G protein-coupled estrogen receptor (GPER) [44]. The researchers generated a 3D pharmacophore model derived from the binding pockets of both agonists and antagonists at the GPER and screened for molecules with similar pharmacophoric features. After the first round of VS and visual inspection, the selected hits were docked into the GPER homology models of the active and inactive states, respectively. One compound was found to be selective for GPER over the α and β estrogen receptors and to inhibit breast cancer cell proliferation.

#### 2.1.7. Histamine H_3_ Receptor

The histamine H_3_ receptor (H_3_R) is activated by the biogenic amine histamine. Multiple central nervous system (CNS)-related disorders, including Alzheimer’s disease and schizophrenia, are affected by intracellular histamine levels, prompting efforts to identify H_3_R inhibitors [72]. Ghamari and coworkers combined structure-based and ligand-based approaches to identify H_3_R antagonists [32]. The group used the X-ray crystal structure of the muscarinic acetylcholine receptor M_3_ (M_3_R) as a template to generate a homology model of the H_3_R. They then generated a structure-based 3D pharmacophore from the H_3_R model in complex with the clinically approved antagonist pitolisant. A focused library of H_3_R antagonist-like structures was generated by performing a ligand-based screening of the ZINC database [73] using pitolisant and identifying compounds with 50% chemical similarity as defined by the Tanimoto similarity index [74]. This library was screened using the 3D pharmacophore, and hit compounds were scored by their ability to fulfill the features of the query pharmacophore, then filtered in-silico according to drug-likeness, pharmacokinetic profiles and potential toxicity. The remaining compound was tested for its affinity at the human H_3_R by a displacement assay using radiolabeled [^3^H]-Nα-methylhistamine, showing H_3_R affinity in the micromolar to submicromolar range. This compound can now be utilized for the development of novel CNS-selective H_3_R antagonists [32].

Frandsen et al. [75] constructed a 3D pharmacophore for the H_3_R orthosteric binding site using a fragment-based method and a reference library of moiety-residue pairs from X-ray crystal structure fragments, which is available through the GPCRdb [28,76]. The resulting 3D pharmacophore was used in VS, generating 44 hits, of which four were identified as having affinity and potency in the micromolar range via pharmacological evaluation. Subsequent pharmacological assaying of a series of analogues of the initial four active hits identified four further antagonists. Of these hits, six were finally identified as truly active; five antagonists and one inverse agonist. Of these, three were not only structurally distinct from the other three ligands and from known H_3_R ligands, but also amongst each other.

#### 2.1.8. Melanin-Concentrating Hormone Receptor-1

The melanin concentrating hormone receptor-1 (MCH-R_1_) is a promising target in the treatment of obesity [77,78]. In their search for MCH-R_1_ antagonists, Helal and co-workers applied PBVS of a commercial compound database to discover nine compounds that showed a displacement of the radiolabeled endogenous ligand MCH by more than 50% at a concentration of 20 μM [40]. Two ligand-based pharmacophore models based on reported MCH-R_1_ antagonists were developed and validated for their predictive ability using a different set of MCH-R1 antagonists. Hit compounds underwent visual inspection with respect to their toxicity and chemical stability and subsequent biological evaluation. We were thus able to identify novel MCH-R1 antagonists eligible for further optimization and use in the treatment of obesity [40].

#### 2.1.9. Neurotensin Receptor Type 1

In their search for antagonists against the class A GPCR neurotensin receptor type 1 (NTR1), Zhang et al. conducted a PBVS. The authors extracted compounds with reported NTR1 antagonistic activity from the BindingDB [79] to generate a ligand-based 3D pharmacophore, which was used to screen a focused library of NTR1 ligands. The resulting hits were tested for their biological effect on NRT1 using a calcium flux assay, identifying one compound with micromolar NRT1 affinity and significant NRT1 inhibitory effects [49].

#### 2.1.10. Protease-Activated Receptor 2

The protease-activated receptor 2 (PAR2) is implicated in pain as well as inflammation, prompting investigations into novel PAR2 antagonists. Cho et al. constructed a shared-feature pharmacophore from a set of experimentally determined PAR2 antagonists and used this in VS of commercially available compound databases [45]. Hits were filtered in silico by drug-likeness and physicochemical properties. Experimental testing of the remaining compounds using a Ca^2+^ mobilization assay, a cytotoxicity assay and measurement of nitric oxide levels in CHO cells overexpressing PAR2 identified hits displaying PAR2 antagonism and anti-inflammatory effects.

### 2.2. Medicinal Plants in Virtual Screening

For many centuries, medicinal plants have been used in traditional medicine, and the herbal ingredients of medicinal plants provide a rich source of potential hits in VS.

#### 2.2.1. TGR5

The Farnesoid X receptor (FXR) is a bile receptor belonging to the nuclear receptor subfamily and is highly expressed in the liver, kidney and adrenal gland [80]. Activation of FXR is an effective treatment of hyperlipidemia [81,82].The FXR agonist 6-ethyl-CDCA, known as obeticholic acid (OCA), additionally activates the GPBAR1/TGR5 [83] and is responsible for unwanted effects of synthetic FXR ligands such as itching. For the discovery of selective FXR agonists, Chen and coworkers used three approaches to the VS of the Traditional Chinese Medicine Database (TCMD, version 2009) for selective FXR agonists [30]. Firstly, a ligand-based 3D pharmacophore was derived from known FXR agonists. Secondly, molecular docking was performed, and the compounds were ranked by their docking scores to evaluate their fit into the FXR binding site. Thirdly, a ligand-based 3D pharmacophore was constructed from known GPBAR1 agonists or the screening of selective FXR agonists with no GPBAR1 activity. Only compounds that matched the FXR pharmacophore, scored well in molecular docking and did not match the GPBAR1 pharmacophore were selected for further investigation. Bioactivity of these hits was detected in HepG2 cells and MD simulations were performed for the elucidation of interactions between the receptor and the ligand. This research displays the use of natural products and provides a new approach to design novel selective FXR agonists.

#### 2.2.2. µ-Opioid Receptor

Morphinans are a class of opioid drugs clinically used to manage severe pain by targeting opioid receptors (OR) [84,85,86]. Kaserer and coworkers [34] performed molecular docking of known agonists, antagonists, and inactive compounds into the µ opioid receptor (µOR) binding site to analyze the important protein–ligand interactions responsible for µOR activity before 3D pharmacophore generation and subsequent VS. The differences in receptor–ligand patterns between the ligand classes indicated that certain interactions (namely, a charged interaction to D147^3.32^ and a hydrogen bond to Y149^3.33^) are necessary for ligand binding to the µOR, and additional pharmacophore features determine whether the ligand acts as an agonist or an antagonist [34]. Subsequently, they generated five ligand-based 3D pharmacophores based on a subset of the initial dataset, and one structure-based 3D pharmacophore based on the inactive µOR crystal structure (PDB-ID: 4DKL [87]). Out of these 3D pharmacophores three represent agonist pharmacophores while the remaining three represent antagonist pharmacophores. The top four ranked hits from the agonist screening and the top two hits from the antagonist screening were experimentally tested for their binding to µOR using competitive binding assays at the µOR and for their analgesic effects using thermal and chemical nociception assays, with one compound displaying µOR antagonist activity. These results in turn facilitated the design of new µOR agonists, as discussed in Section 2.3.1 [35]. Due to high sequence identities between the opioid receptor subfamilies several ligands can modulate either subsets or all of the opioid receptors. One of the aforementioned found hits, acting as a µOR antagonist, was later found to be a novel scaffold κ-opioid receptor (κOR) antagonist and was evaluated in in vitro, in vivo and in silico experiments [88] as κOR antagonists act as promising new treatments for certain mood disorders [89].

### 2.3. Biased Ligands

GPCRs can recruit different signaling proteins after ligand-dependent activation, such as G proteins and arrestins [90,91]. Each allosteric coupling process of the ternary complex of ligand, receptor and transducer results in the activation of distinct downstream signaling pathways, causing different cellular effects [6,91,92]. Ligands that activate downstream signaling cascades to different extents are called ‘biased’ or ‘functionally selective’. The concept of functional selectivity gained a lot of attention as, for some targets, the desired effects are mediated through a signaling pathway that is different to one that leads to the unwanted side effects. Biased ligands that only or predominantly activate the desired pathway are likely to exhibit an improved side effect profile along with retaining their therapeutic effect [61,62,63]. The complexity of ligand-dependent bias is demonstrated by the sustained research into appropriate quantification systems since it must be segregated from system bias, and reproducibility must be ensured by avoiding observational bias. System bias takes all molecules involved in a signaling process into account except the ligand. Recently published community guidelines by Kolb et al. propose the differentiation of ligand bias into benchmark, pathway, or physiology bias, depending on the reference ligand [91,93,94]. For instance, in ligand physiology bias, the endogenous ligand of a receptor is usually defined as ‘unbiased’, meaning that its ratio in activating the different pathways is set as the standard for this receptor [91,93]. To quantify ligand pathway bias, a pathway-balanced ligand can be chosen as reference.

#### 2.3.1. Opioid Receptors

In the field of OR agonists in particular, many efforts have been undertaken to design biased agonists, as G protein-biased ligands have been proposed to act as potent analgesics with reduced side effects [95]. Kaserer and coworkers [35] performed a PBVS against the human µOR by using the six 3D query pharmacophore models described in Section 2.2.2 [34] in parallel to the screening of an in-house library comprising naturally occurring alkaloids and their synthetic derivatives [35]. Five of the models were generated from the chemical structures of a set of experimentally tested µOR agonists and antagonists, and one was derived from a ligand–target complex. The screening process resulted in 15 hit compounds from which eight compounds were chosen for experimental investigation.

Firstly, binding to the orthosteric µOR binding pocket was established via a competitive radioligand binding assay using the known µOR agonist [^3^H]DAMGO. As the compounds corydine and corydaline showed the highest affinities, their recruitment profiles of downstream mediators were assessed via a GTPγS assay and the PathHunter β-arrestin2 recruitment assay. The [^35^S]GTPγS assay evaluates the capability of a ligand–receptor complex to induce G protein recruitment, and the PathHunter β-arrestin2 recruitment assay measures β-arrestin2 recruitment. Both compounds showed full MOR agonism in the [^35^S]GTPγS assay, but no β-arrestin2 recruitment was detectable for either compound. Thus, corydine and corydaline exhibit G protein bias at the µOR. The two biased µOR agonists were finally tested in vivo for their effect against visceral pain and were found to induce antinociceptive effects in mice. In agreement with the idea that G protein-biased µOR agonists can show improved unwanted-effect profiles, neither sedation nor major alterations in locomotor activity were observed in mice at the concentrations tested. The newly discovered biased ligands corydine and corydaline therefore represent valuable starting points for further optimization and the development of new, safer analgesics. The complete VS workflow is depicted in Figure 3.

### 2.4. Allosteric Modulators

Allosteric ligands bind to a binding site distinct from the orthosteric endogenous ligand binding site, influencing the conformation of the receptor to modulate its affinity for orthosteric ligands. Positive allosteric modulators (PAMs) increase orthosteric ligand affinity, negative allosteric modulators (NAMs) decrease affinity, and silent allosteric modulators (SAMs) exert no effect on the orthosteric binding site themselves but act as competitive antagonists for other allosteric ligands. As GPCR orthosteric sites show higher evolutionary sequence conservation than other receptor regions, allosteric sites provide opportunities for ligand selectivity, especially among subtypes of a receptor family that bind the same endogenous ligand. Allosteric modulators reduce the risk of unwanted effects, along with enabling the fine-tuning of receptor activity and downstream signaling, and reducing the risk of potential overdose. Reviews such as those by Wu and colleagues [96] and Wold and coworkers [97] provide a comprehensive overview of GPCR allosteric modulation.

#### 2.4.1. Chemokine Receptors

The chemokine (CC) receptor family consists of 19 class A GPCRs [98], and ligands capable of inhibiting CCR5 activity are popular candidates in treating cancer [99]. El-Zohairy and coworkers [31] based their investigations on the X-ray crystal structure of CCR5 bound to the NAM Maraviroc (PDB ID: 4MBS [100]). They retrieved compounds with experimentally known CCR5 inhibitory activity from the BindingDB [79] to generate several ligand-based pharmacophore models. These were validated by their ability to distinguish between known actives and inactives at the CCR5 allosteric site, and the best-performing pharmacophore model was used in VS. Molecular docking was applied for the binding mode examination of the obtained VS hits. The generated poses were filtered according to their protein–ligand interaction fingerprint (PLIFs) and the top-scoring compounds underwent biological evaluation. Three compounds showed activity against colorectal cancer cells equivalent to or higher than that of Maraviroc.

#### 2.4.2. Metabotropic Glutamate Receptor 1

The class C metabotropic glutamate receptor 1 (mGluR1) is a popular target in the treatment of neuropathic pain. Jang et al. [39] generated a set of known mGluR1 NAMs and clustered them by chemical similarity. A shared-feature 3D pharmacophore was generated from each cluster, and the models were ranked by their ability to identify true active compounds from a separate set of mGluR1 NAMs. The most successful model was used in PBVS. The resulting hits were further filtered using an ML model, which is described in further detail in Section 6. Finally, the hits were docked into the allosteric site of an mGluR1 homology model and filtered by Lipinksi and ADME/Tox filters. The remaining 35 compounds were tested experimentally for their ability to inhibit mGluR1 activity. This study resulted in the identification of four NAMs that are structurally distinct to existing mGluR1 antagonists [39].

## 3. Structure-Activity Relationships and QSAR

### 3.1. Parathyroid Hormone-1 Receptor

Zhong and coworkers [101] generated a representative 3D pharmacophore agonist model from a set of parathyroid hormone-1 receptor (PTHR1) agonists, and a representative antagonist 3D pharmacophore from a set of antagonists. Comparing the two models, they surmised that an extra hydrogen bond feature in the agonist-derived 3D pharmacophore determines the activity of PTHR1 ligands as agonists or antagonists. The authors retrieved two compounds from a virtual database that share a parent chemical structure but differ in their hydrogen bonding capabilities. They tested the activities of the compounds at the PTHR1 via a calcium flux assay, determining that CPU01 displays antagonistic activity at the PTHR1, and CPU03 displays agonistic activity. CPU01 mapped to the antagonist-based 3D pharmacophore and CPU03 mapped to the agonist-based 3D pharmacophore. CPU03 showed much greater affinity to PTHR1 than CPU01, which was attributed to the fact that CPU03 fulfilled the agonist-based 3D pharmacophore model features to a greater extent than CPU01 was able to fulfill the antagonist-based model. Thus, these findings demonstrate the ability of 3D pharmacophore modeling to rationalize SARs by defining characteristics of PTHR1 agonists and antagonists [101].

### 3.2. Dopamine D_2_ Receptor

The majority of orthosteric dopamine D_2_ receptor (D_2_R) ligands contain a protonatable nitrogen that is a key element of the classical pharmacophore model and the model constructed by Ekhteiari Salmas et al. [102]. The protonatable nitrogen is able to interact with the conserved residue Asp^3.32^, a proposed key anchor for basic moieties of aminergic ligands [103,104,105]. Xiao et al. obtained D_2_R antagonists that did not possess a basic protonatable nitrogen [106]. The compounds were utilized by Kaczor et al. [107] to perform molecular docking to a homology model of the inactive conformation of D_2_R, based on the crystal structure of the dopamine D_3_ receptor (D_3_R) (PDB ID: 3PBL) [108]. The molecular docking of the compounds was followed by MD simulations. The compounds were used to create 3D-QSAR comparative molecular field analysis (CoMFA) models, whereby the compounds are placed into a grid and their interactions with other molecules are calculated. The created models were validated by an external test set of compounds.

### 3.3. Serotonin 5-HT_7_ Receptor

Kelemen et al. [109] applied a pharmacophore, proposed by Lopez-Rodriguez et al. [110,111] and modified by Medina et al. [112] consisting of three hydrophobic regions (HYD), one positive ionizable atom (PI) and an H-bonding acceptor group (HBA) to design Spiro[pyrrolidine-3,3′-oxindoles] as selective 5-HT_7_R ligands. Based on this pharmacophore, multiple derivatives of Spiro[pyrrolidine-3,3′-oxindoles] with varying linker lengths, substitution patterns and connectivity types were synthesized and biologically evaluated, which allowed the team to perform an SAR analysis of the compounds [109].

### 3.4. TGR5

Sindhu et al. [54] used a novel compound series of 5-phenoxy-1,3-dimethyl-1H-pyrazole-4-carboxamides, which were identified as TGR5 agonists to generate pharmacophores. The three best-performing pharmacophores as scored by the PHASE program [113] were selected for 3D-QSAR development. The partial least square regression (PLS) factors that allow projection of the activity of compounds and the predictivity of each of the three best pharmacophore hypotheses were analyzed with a test set, which was aligned with the hypotheses. The best post-validation hypothesis was utilized for the creation of contour maps, which depict favorable and unfavorable regions regarding the activity of the compounds for specific properties, such as electron withdrawing and hydrophobic features of the compounds. The analysis of the contour maps was used to determine features for the interaction between the ligand and the receptor. Molecular docking was performed to predict the binding mode of the novel TGR5 agonists. These contour maps can now be used to develop novel and more potent TGR5 agonists.

### 3.5. G Protein-Coupled Receptor 40

Nath et al. [50] employed a dataset of compounds featuring a 3-aryl-3-ethoxy-propanonic acid scaffold and displaying G protein-coupled receptor 40 (GPR40) agonistic activity for the development of pharmacophore hypotheses. The pharmacophore hypotheses were validated by two different methods. The first method of validation was based on survival score. For the second method, a training set and a test set of compounds were used for 3D QSAR generation, which was used for the validation of the pharmacophore. The pharmacophore was used for PBVS alongside structure-based VS. Common hits between the screenings were further inspected and the top five hits were selected based on multiple criteria, including pharmacophore fit score, key interactions, lipophilicity and favorable blood-brain barrier (BBB) penetrability. MD simulations were performed with one of the five best compounds for further studies. Hence, the application of 3D-QSAR and pharmacophores allowed the generation of a robust workflow for the discovery of novel drugs.

## 4. Scaffold Hopping and Hit-to-Lead Optimization

### 4.1. Histamine H_4_ Receptor

The histamine H_4_ receptor (H_4_R) has been implicated in atopic dermatitis (AD), spurring interest in the development of H_4_R antagonists. Ko et al. [36] generated eight distinct shared-feature 3D pharmacophore models from known H_4_R antagonists. These 3D pharmacophores were used for VS of a library of structurally diverse chemical structures generated from the ZINC database of commercially available compounds, resulting in 291 hit compounds. One compound showed an IC_50_ value below 10 μM in a competitive radioligand binding assay at the H_4_R, which bore a tricyclic scaffold not featured in previously known H_4_R antagonists. This compound also showed selectivity for H_4_R over the histamine 3 receptor (H_3_R), whose activation is implicated in H_4_R antagonist-induced unwanted effects. Molecular docking of this initial compound into a homology model of the H_4_R based on the template X-ray crystal structure of the histamine H_1_ receptor (PDB ID: 3RZE [114]) allowed structure-based hit-to-lead optimization resulting in the identification of compound with a novel chemical scaffold which showed an improved inhibition and selectivity profile compared to the initial VS hit. The compound of interest also showed significant anti-pruritic and anti-inflammatory effects in a mouse model of AD.

### 4.2. Somatostatin Receptor Subtype-2

Ishida et al. [115] set out to identify nonpeptidic orally available small molecule somatostatin receptor subtype-2 (SSTR2) agonists. The group decided on a quinoline scaffold and synthesized a series of derivatives, testing their SSTR2 agonist activity by an intracellular cAMP concentration measurement assay. A 4-aminopiperidine derivative displayed the highest agonist potency, as measured by EC_50_, and its structure was used as the basis for scaffold hopping. The authors created a ligand-based 3D pharmacophore from the structures of two known SSTR2 agonists and aligned it with the structure of the 4-aminopiperidine derivative hit. They identified a hydrophobic feature of the 3D pharmacophore not occupied by this initial hit compound, prompting them to synthesize a 5-phenyl pyridine derivative. This second compound was able to fulfill the hydrophobic feature and showed a 42-fold increase in potency over the initial hit.

### 4.3. Serotonin 5-HT_2B_ Receptor

Serotonin 5-HT_2B_ receptor (5-HBT_2B_R) antagonists are under investigation for the treatment of migraine, cardiac failure, and pulmonary arterial hypertension. However, clinical approval of 5-HT_2B_R antagonist candidates is hampered partly by insufficient selectivity of the compounds over other 5-HT_2_ receptor subtypes. Gabr et al. [116] used a modeled complex of the 5-HT_2B_R (PDB ID: 4IB4 [117]) in complex with the known 5-HT_2_ receptor family antagonist doxepin to generate a ligand-induced model of the receptor inactive state [118]. From this, the authors generated a structure-based 3D pharmacophore of the inactive 5-HT_2B_R orthosteric binding site. Mapping of this 3D pharmacophore to the chemical structure of a known 5-HT_2B_R antagonist [119] guided the synthesis of a compound featuring a novel biphenyl amide–tryptamine hybrid scaffold that showed greater fulfilment of the 3D pharmacophore features than the known antagonist. The authors synthesized a series of derivatives featuring this new scaffold, testing their 5-HT_2B_R inhibition via cellular functional assays and their selectivity by measuring antagonistic activity against seven other 5-HTR subtypes. This elucidated the potency and selectivity determinants of the novel compound series, which will be useful in guiding future design of 5-HT_2B_R-selective antagonists.

### 4.4. G Protein-Coupled Receptor 139

Shehata and colleagues [120] had previously generated a ligand-based 3D pharmacophore model from known G protein-coupled receptor 139 (GPR139) agonists. They used this model for VS and assayed 12 of the resulting hits for GPR139 agonist activity. This resulted in the identification of 12 novel GPR139 agonists, featuring novel aromatic bioisosteres and differing linker lengths compared to previously published agonists. They also worked with three distinct series of structurally related known GPR139 agonists. They took all four sets of analogues and generated SARs to determine the moieties responsible for GPR139 activity. The SAR of all of the combined agonists was used to refine the original 3D pharmacophore model. The refined 3D pharmacophore showed good retrieval of actives and non-identification of inactives, confirming their selectivity and sensitivity, and matched all the agonists described in the study. Thus, the 3D pharmacophore screening firstly resulted in the identification of structurally novel agonists, and then a refined 3D pharmacophore model was generated, which showed higher specificity and selectivity than the original screening pharmacophore. This 3D pharmacophore model could therefore be used in prospective VS for novel GPR139 agonists.

## 5. Dynamics in GPCR-Based Pharmacophore Modeling

MD simulations describe the motion of molecular systems such as biomolecules or ligand–target complexes in atomic resolution. Their use provides insight into the conformational plasticity and flexibility of biomolecules and ligand–protein complexes, aiding the study of functional protein mechanisms. The analysis of dynamic binding pocket characteristics and the investigation of dynamic ligand interactions via MD simulations plays an important role in the design and optimization of small molecules. Furthermore, MD simulations can be used to elucidate cryptic pockets in target structures, which are not visible in the static structure, but open and increase in volume throughout the course of a simulation [121].

In an MD simulation, interactions between atoms are estimated by a force field, also termed the potential energy function. Each atom is represented as a point in space with mass, charge and van der Waals parameters, and the dynamics of the system are then computed by solving Newton’s equation of motion as a function of time. The potential energy function consists of bonded or covalent terms, which describe bond stretching, bond-angle bending, dihedral-angle torsion as well as non-bonded Coulomb and van der Waals interactions. The major families of fixed charge atomistic force fields are AMBER [122], CHARMM [123], GROMOS [124] and OPLS [125]. These all share the same functional form and are parametrized for biomolecules. During the simulation, the movement and interactions of the atoms are analyzed for a user-defined time, resulting in time-resolved trajectories which represent possible conformations of molecular systems [126]. The extensive open-source software OpenMM [127] provides an MD simulation toolkit, including a tool for the preparation of the molecular system, which allows free development of new simulation protocols or novel functional forms of interactions. It also enables the freely available and user-friendly usage of the major force fields mentioned above, and the combination of file formats of other popular molecular modeling tools [128].

In recent years, it has become possible to run all-atom MD simulations on graphical processing units (GPUs), allowing parallelization of the simulation workload, distributed by the central processing units (CPUs) of a high performing computing system. This development has facilitated the simulation of membrane-embedded proteins such as GPCRs on the microsecond timescale [129]. The GPCRmd database is an open access scientific platform which can be used to visualize, analyze and share GPCR-related MD-derived trajectories. The community project aims at exploring and analyzing key aspects of GPCR dynamics and contains simulations of all known GPCR classes known to date. An array of tools including distance plots, water density maps and interaction networks allows the study of molecular interaction patterns. Furthermore, the GPCRmd database links the simulations to pharmacological or biochemical data such as mutations or electron density maps of X-ray structures. This feature enables the elucidation of mechanistic models of GPCRs and related diseases. A protocol for common system preparation is available on Github [130].

Together with the quickly growing number of structural GPCR data, 3D pharmacophores derived from and refined by MD simulations could be a quantum leap in providing deeper and refined insights into GPCR pharmacology and drug design. This section will highlight the advances of the combination of these two techniques in the GPCR field.

### 5.1. Dynamic Pharmacophores

The dynamic pharmacophore, or dynophore, method allows the unique combination of 3D pharmacophore models with MD simulations (Figure 4A). Dynophore is a fully automated application which generates chemical feature-based interaction patterns of the ligand and its interacting residues throughout MD trajectories [15,131]. Single protein–ligand interaction points generated sequentially for every timestep of the MD simulation are then grouped into so-called ‘superfeatures’ which are represented by point density clouds. This method allows statistical analysis of ligand–target interaction frequency and occurrence and provides an easily accessible dynamic view of interactions. The dynophore method has been successfully applied to GPCRs to describe complex phenomena of GPCR pharmacology such as partial agonism and has helped to elucidate ligand-dependent mechanistic understanding of GPCR function [92,132,133]. Complicated signaling events such as biased agonism can be difficult to capture via static models, so the incorporation of dynamics into 3D pharmacophores afforded by the dynophore method allows a deeper understanding of how pharmacophore interaction patterns relate to downstream signaling events.

#### 5.1.1. M_1_ Receptor

Volpato and co-workers used the dynophore method to investigate dynamic interaction patterns of the potent bitopic ligand 5-C8 which simultaneously targets the orthosteric and allosteric binding sites of the muscarinic M_1_ receptor (M_1_R). As the allosteric binding site is less conserved than orthosteric acetylcholine binding site within muscarinergic GPCRs, it is of interest in the design of M_1_ subtype-selective ligands. A toolbox of benzyl quinolone carboxylic acid derivatives (BQCAd) with diverse orthosteric building blocks was pharmacologically evaluated for their potency and efficacy at the allosteric and orthosteric sites in terms of G protein signaling and β-arrestin recruitment. The dynophore method allowed the analysis of the residues interacting with 5-C8 within both binding sites, highlighting Y179 and W400^7.35^ within the allosteric vestibule as key interacting residues for the BQCAd moiety. This study provides a blueprint for the mechanistic understanding of allosteric modulation at the M_1_R, paving the way for M_1_R subtype-selective ligand design [133].

#### 5.1.2. 5-HT_2B_ Receptor

The serotonergic receptor 5-HT_2B_R served as a model system to gain detailed mechanistic understanding of ligand-dependent biased signaling via MD simulations and dynophores. The study revealed that conformational interference of the β-arrestin biased ligands ergotamine and lysergic acid diethylamide within the extracellular vestibule restricts the signaling repertoire of the 5-HT_2B_R in a manner relating the degree of ligand bias to the degree of closure of the extracellular loop region. Dynophores of serotonin, lysergic acid diethylamide and ergotamine revealed common as well as distinct interaction patterns of the three ligands, revealing specific conformations of biased ligands within the 5-HT_2B_R. This study also highlights the importance of considering distinct ligand-bound receptor conformations during VS campaigns aiming to find hits for biased GPCR ligands at aminergic receptors [133].

### 5.2. PyRod

The free and open-source software PyRod has also recently been developed by Schaller et al. [134]. PyRod analyzes the movement of water during MD simulation with respect to the protein environment, since ligands usually compete with water molecules within binding sites for binding events (Figure 4B). The information derived from these interaction analyses is then represented in so-called dynamic molecular interaction fields (DMIFs) which can then be translated into corresponding 3D pharmacophore features. These can be applied to prospective VS campaigns. Since no ligand information is necessary to derive the 3D pharmacophores, PyRod is a promising tool to identify new drugs for orphan GPCRs. Supported chemical feature types comprise hydrogen bonds, ionizable and hydrophobic features, and aromatic interactions. PyRod was applied to the MCHR1 receptor for a retrospective screening of active ligands. The generated 3D pharmacophores performed better in discriminating active ligands compared to ligand-based pharmacophores since they incorporated information on protein structure as well as dynamics [135].

### 5.3. Dopamine D_2_/D_3_ Receptor

Ferruz and coworkers used a combination of molecular dynamics and Markov state models to investigate the dynamic interaction patterns and binding modes of the dual dopamine D_2_R/D_3_R antagonists GSK598809 and PF-4363467. They compared these to the interactions of the known antagonists haloperidol and eticlopride bound to D_3_R after performing molecular docking of the ligands. PF-4363467 shows a different 3D pharmacophore feature arrangement compared to the other three ligands. Each binding hypothesis was rationalized via point mutation studies, measurement of binding affinity and radioligand binding assays. Extensive MD simulations revealed a novel binding pose for PF-4363467 as well as a cryptic pocket which opened up through the displacement of F346^6.52^, that is situated between helices V and VI [121].

Besides the aforementioned applications, MD simulations have been combined with PBVS to establish the stability of agonist-GPR40 complexes as well as ligands at the D_2_R obtained via in silico screening (101, 108). Furthermore, MD simulations are involved in binding free energy calculations, such as the MM-PBSA/GBSA method which has also been applied after 3D pharmacophore or docking-based VS campaigns [32,37].

## 6. Machine Learning and 3D Pharmacophore Models in GPCR Drug Discovery

3D pharmacophore and ML methods [136,137,138] are widely used in the GPCR drug discovery community. In this review we have shown that 3D pharmacophores can be utilized at different stages of the drug discovery pipeline for GPCRs, e.g., in SBDD and LBDD strategies as a VS method (Section 2), guiding the selection of plausible molecular docking results, or in SAR studies to highlight interaction patterns important for affinity to the target. These different stages in GPCR drug discovery, especially the prediction of bioactive ligands (VS), can also be performed with ML methods, as has been reviewed several times [136,137,138]. Furthermore, methods, such as HS-pharm [139], Pharm-IF [10], and DeepSite [140], that combine ML with 3D pharmacophore methods have been briefly discussed in a review on 3D-pharmacophore modeling by Schaller et al. [15]. This section will highlight the combined use of machine learning methods with pharmacophores to elevate GPCR drug discovery.

Beyond pharmacophore-based approaches, state-of-the-art ligand-based predictions utilize general molecular properties of ligands (e.g., molecular weight, log P, and topological polar surface area), graph-based methods, and fingerprints (Figure 5) representing ligand information in a machine readable form [136]. In order to be comparable, molecular property data from different sources must first be normalized to standard activity types and units. To this end, the Chembl database features a p-chembl value, defined as: −log10 (molar IC50, XC50, EC50, AC50, K_i_, K_D_ or potency) [141]. The p-chembl value allows the comparison of approximately standardized measures of ligand potency and affinity. When implemented in ligand-based ML methods, pharmacophores are represented as fingerprints [10,142,143,144,145]. Each of these pharmacophore-based fingerprints differ in their method of encoding the relationship between pharmacophore feature points of a ligand. Two-point pharmacophore fingerprints (e.g., Chemaxon PF fingerprint [142]) encode ligands based on combinations of two feature points with their measured pairwise topological distance (as space or molecule bonds) (Figure 5). Three-point pharmacophores describe three features in a triangular relationship connected through three topological distances between the features (e.g., Gobbi2D [143], 2D-FTP [144]), while 4-point fingerprints (Pharm-IF) consist of four feature points connected via six inter-feature distances (Figure 5). Distances are normally encoded as defined ranges leading to less constrained models. The use of 4-point fingerprints can be more computationally demanding compared with 2- and 3-point fingerprints due to the larger binary bit length needed to encode 4-point fingerprints [10,145].

A new pharmacophoric fingerprint (Pharmacoprint) has been developed recently by Warszycki et al. utilizing a combinatorial 2- and 3-point fingerprint [145]. To evaluate the discriminative performance between active and inactive ligands of Pharmacoprint, Warszycki et al. collected ligand datasets for 15 targets, including serotonin 5-HT_2A/2C/6_, dopamine D_2_, and µ/δ/κ-opioid receptors, and compared the prediction performance to that of 11 other fingerprints (e.g., Estate [146], MACCS [147], PubChem [148], ChemAxon PF [142], extended-connectivity fingerprint diameter 4 (ECFP4) [149], and functional-connectivity fingerprint diameter 4 (FCFP4 [149]) using three different supervised ML classification algorithms; support vector machine (SVM), linear SVM, and logistic regression. Each dataset consists of true actives and inactives retrieved from the ChEMBL database as well as putative inactives from the ZINC library. Based on the calculated average and median Mathew correlation coefficient (MCC) of the ML models trained with only true active and inactives, Pharmacoprint outperformed all other fingerprints with an average MCC of 0.736 and a median MCC of 0.766 (range of metric −1 to 1, in which 1 is the best performance value). It is worth mentioning that the ECFP4 (average MCC 0.729 and median MCC 0.754) and FCFP4 (average MCC 0.722 and median MCC 0.749) performed similarly to Pharmacoprint.

The authors also revealed a more detailed comparison between Pharmacoprint and the ChemAxon PF fingerprint. Pharmacoprint models outperformed all probed GPCR datasets with every used ML algorithm with an MCC difference ranging between 0.010 and 0.241. Only 5-HT_2A_ and 5-HT_2C_ in combination with SVM showed a similar prediction performance using the ChemAxon PF (MCC difference of −0.001 and −0.008, respectively). Further ML optimization with neural networks, feature reduction/engineering methods, and ligand preparation yielded an MCC of up to 0.962. Even though these results are solely retrospective, the Pharmacoprint fingerprint might show a reasonable performance enhancement for ligand- and pharmacophore-based ML methods in GPCR drug discovery pipelines.

The utilization of structural information for 3D pharmacophore and ML-based GPCR drug discovery inherits the same challenges as for classical SBDD without ML (Section 7). Another interesting approach, besides the direct use of pharmacophore data for ML models, is the hierarchical implementation of VS via ML methods and 3D pharmacophore-based VS as described in the following case studies.

Jang et al. identified novel mGlu1R receptor negative allosteric modulators by applying ligand-based pharmacophore models, Naïve Bayesian ML and structure-based methods in their VS campaign [39]. A performance comparison of standalone and different combinations of each VS step revealed that the hierarchical VS combining all steps had the lowest yield (81 compounds, 5.66%) with the highest hit rate (3.70%).

Two recent studies from the lab of Jianping Lin also implemented a multi-stage VS method using ML, pharmacophore models and molecular docking for the discovery of novel adenosine A_1_ receptor (A_1_R) antagonists [150] and novel dual adenosine A_1_/A_2A_ receptor (A_1_R/A_2A_R) antagonists [151]. The first study by Wei et al. [150] trained three random forest ML classification models with three datasets of known actives and inactives with different K_i_ thresholds (actives thresholds K_i_ < 20 nM or K_i_ < 100 nM, inactives thresholds K_i_ > 200 nM, K_i_ > 1000 nM, and K_i_ > 2000 nM). The model with an actives threshold of <20 nM and an inactives threshold of >2000 nM yielded the best performance and was subsequently used for further model optimization with feature extraction methods, which reduced the number of features from 484 to 31 while enhancing the initial model performance. The optimized model was then used as the first VS step filtering the ChemDiv library with 1,492,362 compounds yielding 141,916 compounds. Subsequent VS steps include the use of a combined ligand- and structure-based pharmacophore model (e-Pharmacophore) validated with a set of 37 known A_1_R antagonists and 1332 generated decoys based on the structures of the known antagonists followed by molecular docking for screening, clustering compounds based on the chemotype, and visual inspection. Finally, 22 compounds were selected and experimentally tested, of which 18 showed binding affinity in a radioligand binding assay leading to a hit rate of 82%. Six of the 18 compounds were further evaluated with a cAMP assay revealing pIC_50_ between 5.51 to 6.38. Three of these six compounds also showed good affinity (pK_i_ 6.11–7.13) while having >100-fold selectivity against A_2A_R.

The second study by Wang et al. [151] utilized two deep learning classification methods (deep neural network (DNN) and convolutional neural network (CNN)) with two fingerprints (ECFP4 and neural fingerprint (NFP)) for feature generation as the first step of the VS campaign. The dataset used to train the deep learning models consists of 310 known active dual A_1_R/A_2A_R antagonists with a threshold of K_i_ < 40 nM and 405 inactive compounds with a threshold of K_i_ > 1000 nM. The DNN and CNN classification models were then used to screen the ChemDiv library (1178506 compounds). Only compounds predicted by both models simultaneously as hits were retained (58,886 compounds). The next VS step utilized 1 out of 11 ligand-based pharmacophore models generated from 14 chemically diverse compounds selected from the dataset of 310 active ligands. To validate the pharmacophore model a set of 42 known dual antagonists (K_i_ < 40 nM) and 913 generated decoys based on the structures of the known actives were used. Further steps in the VS campaign include molecular docking screening and visual inspection. Finally, 19 selected compounds were experimentally evaluated with a cAMP functional assay and radioligand binding assay evaluated for A_1_R and A_2A_R. Eight compounds showed dual antagonistic activity and binding affinity for A_1_R/A_2A_R leading to a hit rate of 42%. From these hits five compounds showed good antagonistic dual A_1_R/A_2A_R activity in the cAMP functional assay (pIC_50_ 4.20–6.78) from which two compounds also showed the highest binding affinity of the 19 tested compounds towards both receptors (pK_i_ 7.16–7.49).

In sum, the combined use of 3D pharmacophore-based models and ML methods can greatly benefit GPCR VS campaigns through information enrichment from the underlying data used, better performance and high hit rates. It must be noted that the prediction methods discussed are primarily used to filter newly designed molecules and are not able to suggest novel chemical scaffolds or substitutions.

## 7. Discussion

### 7.1. Challenges: LBDD Is Favored over SBDD in GPCR Research

As evidenced by the case studies in this review, the research so far on computational GPCR drug design has focused largely on ligand-based over structure-based methods; more studies generated the 3D pharmacophores via ligand-based than structure-based methods (Figure 6). The lack of solved GPCR 3D structures remains a major hurdle in the field. Membrane-bound proteins are particularly difficult to crystallize, and receptor flexibility has further hindered the generation of atomistic structures of GPCRs [152]. This problem is amplified by the significant differences between active, inactive and intermediate receptor states; the 3D pharmacophores of various ligand types (such as antagonist or agonist) bound to their respective receptor structures will differ significantly. Hence, in the absence of an X-ray crystal structure of a receptor in a certain state, it may prove difficult to derive a 3D pharmacophore for retrieval of a desired ligand class via purely structure-based methods.

In the absence of structural information, ligand-based approaches present valuable techniques in 3D pharmacophore modeling. The basic principle of ligand-based methods is that similar compound structures are likely to confer similar properties [157]. The aim of ligand-based pharmacophore modeling is to identify the 3D pattern of chemical features of receptor ligands, which is highly correlated with the input molecule.

A major challenge in the field of LBDD remains the prediction of the bioactive conformation of a ligand from its 2D structure [43]. Thus, the degree of complication involved in 3D pharmacophore generation increases as the number of input ligands and their flexibilities increase. Furthermore, ligand-based methods often rely on bioactivity data deposited in databases—these must be checked for accuracy, and data obtained via different assays or systems are not necessarily comparable [38]. Hence, the conformational search and bioactive data retrieval of small molecules should be carefully considered when preparing ligand libraries.

Interestingly, a 2016 study aiming to identify agonists for the A_2A_R discovered that the ZINC database contained more chemical structures similar to classical adenosine antagonists than agonists. This was postulated to be due to the higher molecular complexity of A_2A_R agonists, which may be the case for agonists of other receptors, such as the P2Y_12_ receptor. The authors also determined a majority of agonist-like compounds for the β2-adrenoceptor and 5-HT_1B_ receptors, which tend to have less complex structures. This led to the conclusion that database bias against molecules may increase in accordance with their molecular complexities [158].

Castleman et al. recently performed an in silico benchmarking study in which they evaluated the performance of various ligand training sets in the generation of 3D pharmacophore models for GPCR ligand discovery [159]. The results indicate that mixed-function ligand sets, consisting of both agonists and antagonists, are the most successful in producing pharmacophore models capable of identifying active compounds. This is most beneficial in the discovery of hits for targets with few known ligands.

As LBDD depends on the structure of the known active compound, it is not optimally suited to the discovery of novel active compound structures. On the contrary, SBDD provides clear opportunities for scaffold hopping and structural diversification. A further advantage of structure-based 3D pharmacophores is their incorporation of exclusion volume spheres, which represent space occupied by the protein. Hit structures overlapping with exclusion volume spheres will be sterically hindered from fitting into the binding site [160]. This constraint can be useful in filtering out false positives whose structures may be identified as hits by a ligand-only pharmacophore model, but do not show shape complementarity within the binding site.

In cases of sparse structural information, in silico modeling methods present valuable opportunities for generating 3D models of GPCRs to fill this gap. In brief, modeling generates a 3D structural model of a receptor based on a template of a known receptor structure with a certain degree of sequence similarity to the target receptor. Proteins within the expansive GPCR family share sequence similarity to various extents, which provides a large number of sequence templates for in-silico modeling methods. The GPCRM [161], GPCRdb [28,76,114] and the GPCR-SSFE [162] are freely accessible online databases for the storage and prediction of GPCR structural models. Historically, hit rates of VS campaigns based on X-ray crystal structures tend to be higher than those of screenings based on homology models [163]. Possible explanations include the unavailability of solved protein structures with sufficient sequence similarity, or inaccuracy introduced by the alignment and structure prediction processes. However, advances in in-silico protein modeling hold promise for addressing these issues.

Threading and ab initio modeling are the methods of choice if the target sequences have low sequence identity compared to their template. Threading involves the alignment of the template sequence to 3D structures of homologous proteins retrieved from the PDB [164]. Ab initio modeling relies on protein prediction conformation from a template sequence guided by energy functions in force fields [165]. Kaushik et al. used ab initio modeling to predict the 3D structure of the G protein-coupled receptor 142 (GPR142) for subsequent ligand-based 3D pharmacophore generation [41].

For template–target pairs with higher sequence identity (empirically >40%), homology modeling is the 3D structure prediction method of choice. Jang et al. constructed a homology model of the mGluR1 based on the X-ray crystal structure of the dopamine D_3_ receptor (D_3_R) (PDB ID: 3PBL [108]). The mGluR_1_ and D_3_R share an active site identity and similarity of 37.5% and 62.5%, respectively. The homology model was used in a hybrid structure-based and ligand-based VS campaign, as described in Section 2.4.2. After five hits had been identified via this method, the X-ray crystal structure of the mGluR1 in complex with the NAM FITM was published (PDB ID: 4OR2 [166]). Comparison of the homology model to the X-ray crystal structure revealed structural differences in the target pocket, resulting in the ligands docked into the homology model adopting different conformations to the co-crystallized NAM. However, ligands in both structures were able to recapitulate these hydrogen bond interactions to T815^7.32 × 33^ and N760^5.47 × 47^ (numbers in superscript denote residue numbers according to the Ballesteros–Weinstein conventional numbering scheme for class A GPCRs [14]) that are reported to be important for mGluR_1_ antagonist binding [39]. Thus, the authors postulated that, although the structure-based studies were performed on an inaccurate receptor homology model, hit compounds were still able to display mGluR_1_ antagonistic activity due to their ability to fulfill the necessary 3D pharmacophore features.

Ligand-guided homology modeling enables structural models to be optimized further through incorporation of knowledge on the binding of existing ligands, such as was done by Lupala et al., who used the structure of a known binder to hBKB1R to guide homology modeling [51]. Guided by information on the necessary receptor–ligand interactions, the authors of these studies were able to select a receptor model capable of fulfilling the necessary interaction patterns with the docked compound into the relevant binding site of the model.

It must be emphasized that the accuracy of input structural models is vital to the success of VS. Studies based on poor quality homology models run an increased risk of retrieving false positive or negative hits. Higher accuracy can be incorporated into models via MD simulations, as performed by Rasaiefar et al. [53], as well as during the investigations into the hBKB1R receptor [51]; the aforementioned receptor–ligand complex was further refined via MD simulation, and the final model was composed of an average of the conformations generated over the final 100 ns of the simulation. Similar methods for ligand-guided homology model generation and refinement via molecular dynamics were carried out on the BkB2R [52], BB1R [53], H_3_R [32], CB2R [33], and H_4_R [36].

Besides the aforementioned in-silico methods, the recent boom in AI-based structure modeling methods, such as AlphaFold [16] and RoseTTAFold [167], represent powerful tools for future use in 3D structure prediction.

In the fortunate case of sufficient available data, it has been proven beneficial to perform both ligand and structure-based techniques, termed a hybrid approach, in a single study for GPCR drug discovery. Such a strategy was applied to the discovery of H_3_R ligands [32], where the synergistic combination of methods allowed mutual compensation for their respective limitations and exploitation of their respective strengths. In their retrospective evaluation of different in silico methods for orthosteric GABA_B_ receptor (GABA_B_R) ligand discovery, Evenseth and colleagues conducted both ligand-based and structure-based pharmacophore VS for the prospective identification of orthosteric GABA_B_R ligands [38].

### 7.2. Evaluating PBVS and DBVS in GPCR Ligand Discovery

When discussing the application of SBDD to the computational investigation of GPCRs, it must be noted that literature definitions of SBDD often merely include docking-based screening methods [168,169], whereas PBVS is usually classified under LBDD. Hence, certain challenges arising during studies classed as SBDD could be overcome via structure-based pharmacophore modeling.

A particular issue concerning the analysis of molecular docking results is the accuracy of the scoring functions used to rank docking poses [152,170,171]. Three-dimensional pharmacophore modeling enables the evaluation of binding hypotheses based on ligand–target interaction patterns, which can be invaluable in prioritizing ligand binding hypotheses based on known obligate interactions. This method was recently employed to filter docking hypotheses of three serotonin receptor agonists at the 5-HT_2B_R in order to rationalize their differing degrees of biased agonism at the receptor [133]. Further examples of this use of 3D pharmacophores in the filtering of putative ligand binding conformations can be found in the studies detailed in Section 2 [30,32,50,101].

In hierarchical VS, PBVS is often used as an initial filter to pare down the number of molecules before molecular docking is performed [31,37,39,42,43,49]. The main reason for this is that PBVS is less computationally expensive than DBVS. Furthermore, PBVS can, to some extent, address the problems arising from the use of inappropriately designed or optimized scoring functions or insufficient consideration of target flexibility in DBVS by introducing a tolerance radius for pharmacological features. The study performed by Jang et al., on the mGluR_1_ is an example of such a hierarchical VS approach [39].

Pharmacophore modeling could avoid false positive VS results obtained via DBVS. Rodriguez et al. determined three receptor–ligand interactions necessary for A_2A_R activation and performed DBVS for A_2A_R agonists using multiple active-state X-ray crystal structures of the A_2A_R [158]. The compounds were ranked by docking score and whether they were able to fulfill at least two of the three receptor-activating polar interactions. Of the nine compounds showing affinity for the receptor, none of them were able to activate the receptor. The authors hypothesized that this could be because none of the compounds were able to fulfill all three interactions, an attribute which could have been defined by screening with a 3D pharmacophore model.

### 7.3. Comparing Different Studies on the Same Receptor

The publication of multiple different models of the same GPCR has seen a variety of computational methods employed in investigations of the same receptor. For example, the 5-HT_2B_R is examined in two different case studies. Gabr et al. [116] generated an antagonist-induced model of the inactive-state receptor (Section 4.3). From this model, they derived an optimal pharmacophore to guide the synthesis of novel antagonists displaying improved potency and selectivity. Such studies are time-efficient and do not require significant computational resources. However, the use of a single, static structure does not allow the insight into receptor flexibility provided by analyses incorporating receptor dynamics. Denzinger et al. [133] investigated three distinct active-state conformations of the 5-HT_2B_R, each in complex with a different biased ligand (Section 5.1.2). MD simulations and dynophore analysis elucidated the ligand–receptor conformations and interaction patterns underlying biased signaling. Although the incorporation of dynamics indisputably allows a deeper understanding of GPCRs, these investigations require more computational power and time. Ultimately, the choice of receptor model and in-silico methods depends on the scope and nature of the research question.

## 8. Future Perspectives

### De-Orphanizing GPCRs

One of the main advantages of 3D pharmacophore modeling is the opportunity to derive a template for VS for the identification of ligands for any apo-state binding site. The independence of structure-based 3D pharmacophore modeling from available ligand data could prove useful in the search for ligands for orphan GPCRs. Structure-based VS could either aid in determining the endogenous ligand, or in identifying pharmacological tool compounds to elucidate the function of orphan GPCRs [163].

The opportunities in GPCR de-orphanization afforded by 3D pharmacophore modeling are exemplified by the study of Pillaiyar et al. on the GPR84 [172]. They constructed a ligand-based 3D pharmacophore model based on known GPR84 ligands, identifying differences in the models between low- and high-potency agonists and thus constructing a 3D pharmacophore describing high-affinity GPR84 surrogate agonists. This could be used in prospective VS to identify further surrogate agonists, or even the endogenous agonist.

For orphan GPCRs whose sequence is known yet whose 3D structure remains unsolved, the structure prediction techniques mentioned in Section 7.1 present viable options for 3D model generation. In their comprehensive overview of structure-based investigations into orphan GPCRs, Ngo et al. present success stories in identifying orphan GPCR ligands using homology models, as well as an iterative workflow for the ligand-guided optimization of homology models of orphan GPCRs [163].

Even at well-studied receptors, binding site prediction tools such as FTMap [147] and FTSite [173] can be used to identify binding sites outside of those already known, even in the absence of co-crystallized ligands. Binding site prediction technology has also been successfully applied to the identification of allosteric GPCR binding sites, amongst them orphan sites, in studies such as that recently described by Hedderich et al. [174].

As scarcity of ligand information and accurate 3D receptor structures present significant obstructions in computational GPCR research; X-ray crystal structures of ligand-GPCR complexes are valuable sources of their associated receptor–ligand interactions. Methods such as Pharmacophore-Map-Pick analyze X-ray crystal structures of GPCR–ligand complexes to facilitate the generation of 3D pharmacophores in the absence of both ligand-based and structural information [46].

## 9. Conclusions

The case studies presented in this review illustrate how 3D pharmacophores serve to bridge the gap between lingering questions in the field of GPCRs and the ever-expanding techniques found in the toolbox of CADD. The power of 3D pharmacophores is multi-faceted and far-reaching; retrospective 3D pharmacophore modeling has been used to rationalize ligand bias and potency, to visualize ligand–target interaction patterns, and to build SARs. Prospectively, 3D pharmacophores serve as starting points for hit-to-lead development and de novo drug discovery through methods including VS and scaffold hopping. Structure- and ligand-based techniques combined allow researchers to tackle historically difficult topics such as de-orphanization. Furthermore, combining 3D pharmacophore modeling with sophisticated methods such as ML and MD simulations allows even deeper mechanistic and pharmacological understanding of GPCRs. Altogether, 3D pharmacophore-guided computational investigations into GPCRs provide a fresh perspective on a long-established field and are well-suited to propel GPCR research along bright new avenues of inquiry.

## Figures and Tables

**Figure 1 pharmaceuticals-15-01304-f001:**
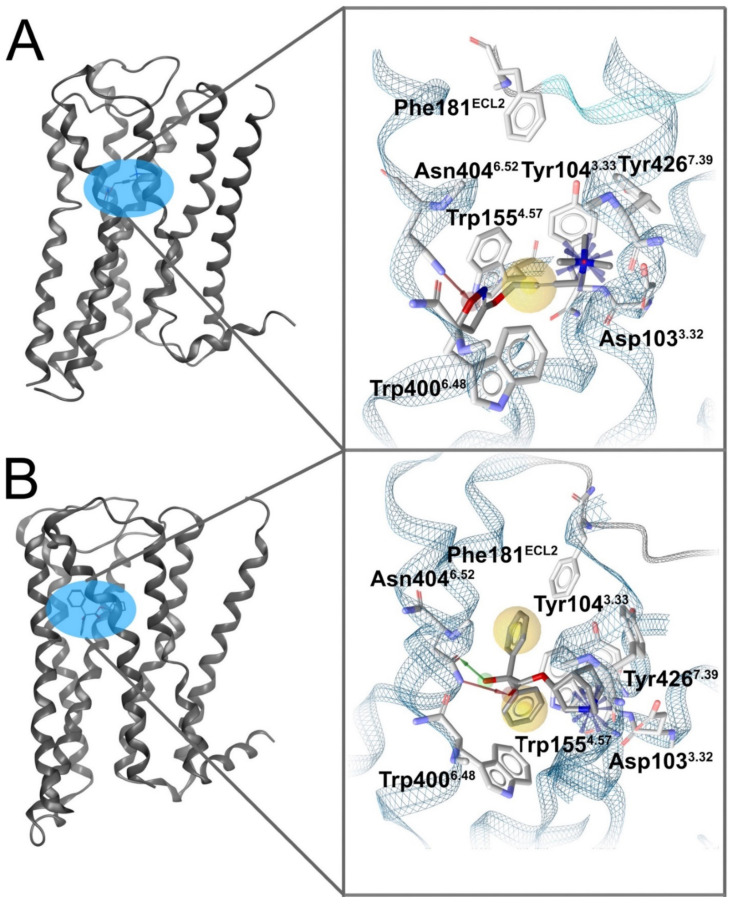
M2 muscarinic acetylcholine receptor in an (**A**) active-like conformation bound to orthosteric agonist iperoxo (adapted from PDB code 4MQS [11]) and (**B**) an inactive-like conformation bound to an orthosteric antagonist (adapted from PDB code 3UON [12]). Yellow spheres = hydrophobic contacts, blue bursts = positive ionizable areas, red arrows = hydrogen bond acceptor positions, green arrows = hydrogen bond donor positions. Binding site conformation representations were created using LigandScout v. 4.4.3 (Inte: ligand, Vienna, Austria) [13]. Residue numbers in superscript follow the Ballesteros–Weinstein numbering system for the conformational numbering of class A GPCR residues [14].

**Figure 2 pharmaceuticals-15-01304-f002:**
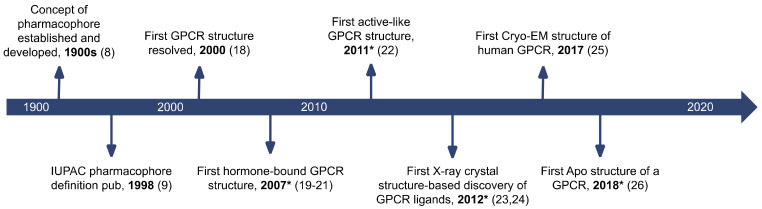
GPCR research milestones related to CADD, * = pharmaceutically relevant.

**Figure 3 pharmaceuticals-15-01304-f003:**
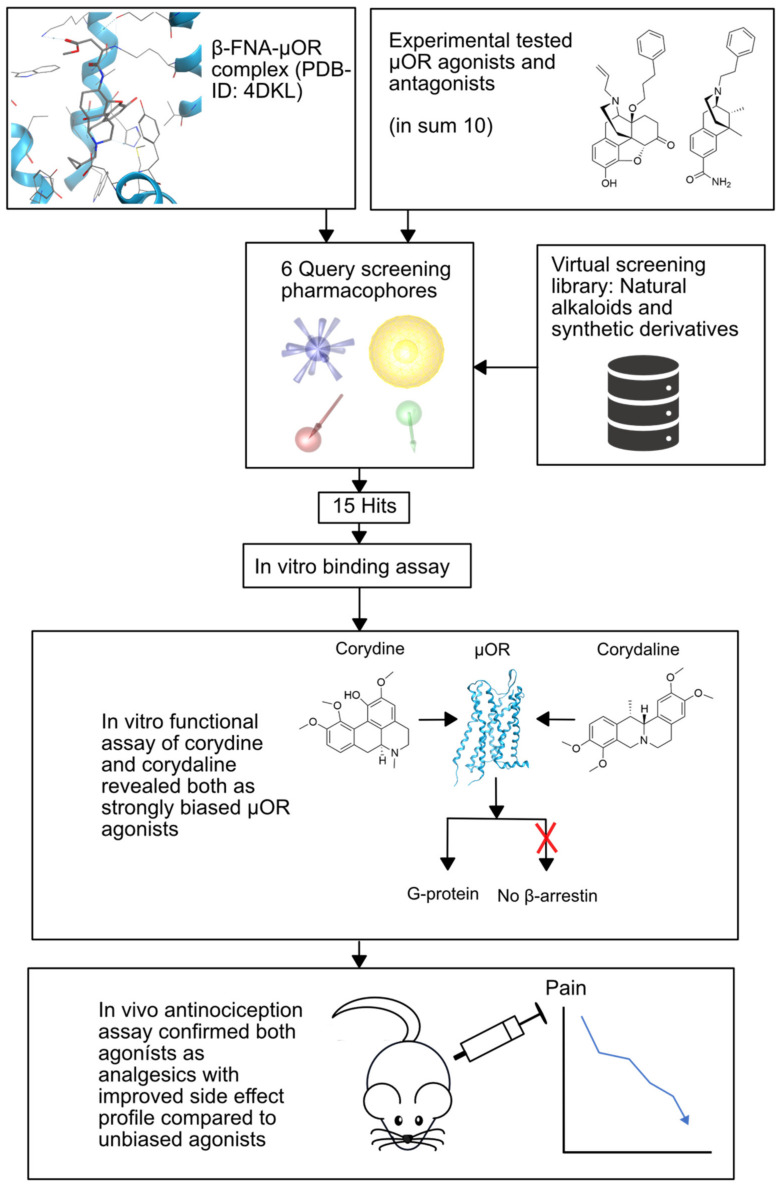
VS workflow carried out by Kaserer et al. [35].

**Figure 4 pharmaceuticals-15-01304-f004:**
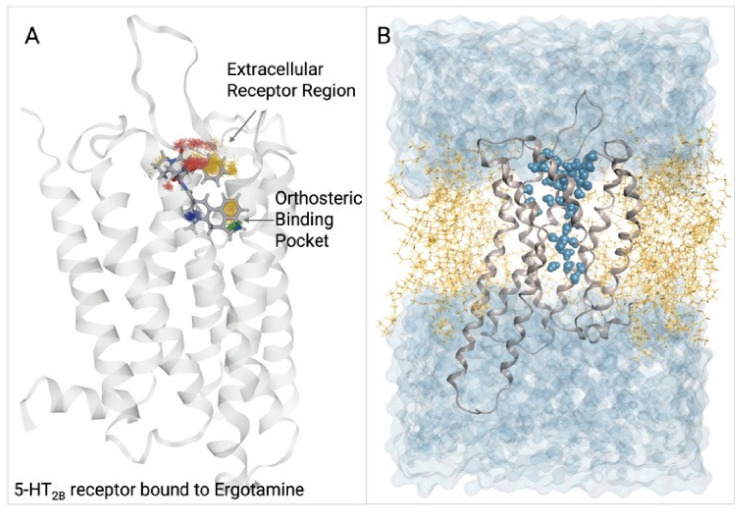
(**A**) Dynamic interaction pattern of the β-arrestin-biased ligand Ergotamine within the 5-HT_2B_ receptor, derived via MD simulation and subsequent application of the dynophore method (https://github.com/wolberlab/dynophores, accessed on 8 July 2022). The interaction pattern indicates a stable binding pose within the orthosteric binding pocket, visualized as spherical distribution of the point density clouds, and a more flexible interaction pattern within the extracellular receptor region [133]. Yellow = lipophilic contact, red = hydrogen bond acceptor, green = hydrogen bond donor, blue = positive ionizable area. (**B**) Time frame snapshot of the 5-HT_2B_ receptor (grey) derived from MD simulation within a box of water molecules (light blue surface) and embedded into a membrane (yellow). The movement of water molecules within the receptor binding site (cyan spheres) during MD simulations can be traced by the open-source software Pyrod [134] which analyzes their interaction patterns to derive 3D pharmacophore features (https://github.com/wolberlab/pyrod, accessed on 24 June 2022).

**Figure 5 pharmaceuticals-15-01304-f005:**
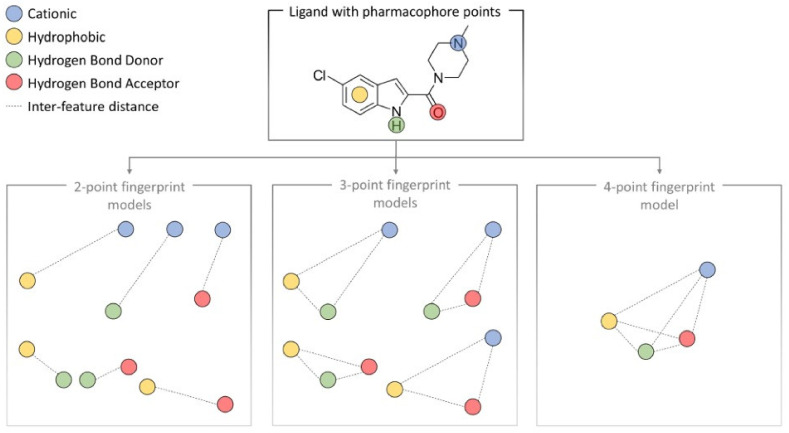
Ligand-based pharmacophore fingerprints encode the relationship between feature points based on the number of features and the measured topological distance between the features.

**Figure 6 pharmaceuticals-15-01304-f006:**
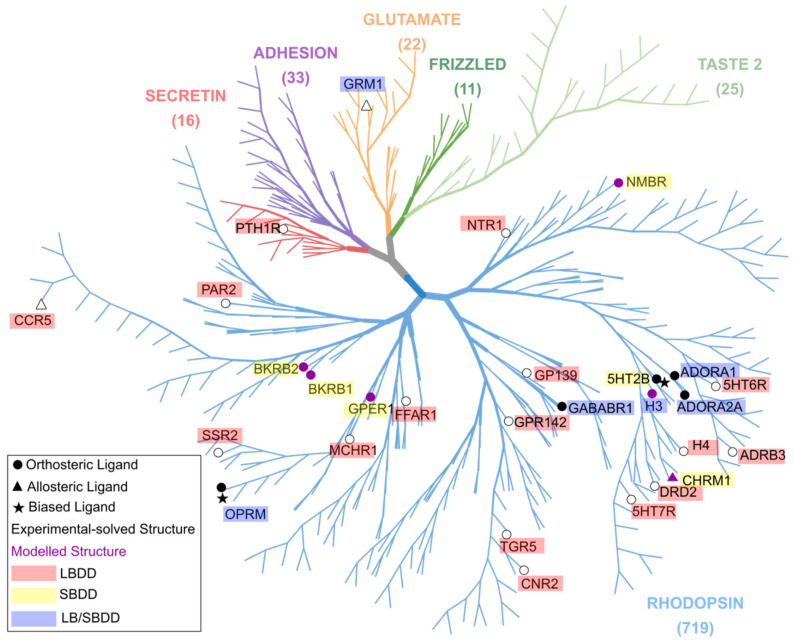
A phylogenetic tree representing all the receptors discussed in this review, and their associated applications of pharmacophore modeling in GPCR drug discovery. Node labels indicate the Uniprot gene name of the receptor and are colored according to the method used to generate the 3D pharmacophore (red, ligand-based; yellow, structure-based; blue, both methods). The shape of the node implies the ligand type studied in each case (circle, orthosteric ligand; triangle, allosteric ligand; star, biased ligand). Circles, triangles, and stars are colored according to the origin of the 3D receptor structure (black, experimentally solved structure; purple, modeled structure; white, no structure involved). The GPCR tree was produced as follows. First, canonical fasta files of all human GPCR receptors (826 receptors in total, but olfactory receptors are not shown on the tree for simplicity and aesthetics) were downloaded from the Uniprot database [153]. Then, the above fasta files were used to generate a phylogenetic tree file by Clustal Omega (default setting) [154]. Finally, the tree was displayed and annotated with iTOL [155]. The tree layout was designed according to [156]. Uniprot gene names of the receptors are as follows: NMBR = Bombesin Receptor 1; BKBRB1 and BKBRB2 = Bradykinin Receptors 1 and 2, respectively; ADRB3 = β3-adrenergic Receptor; CNR2 = Cannabinoid Receptor 2; TGR5 = G Protein-Coupled Bile Acid Receptor; GPER = G Protein-Coupled Estrogen Receptor; H3 = Histamine H_3_ Receptor; MCHR1 = Melatonin-Concentrating Hormone Receptor-1; NTR1 = Neurotensin Receptor type 1; PAR2 = Protease-activated Receptor 2; OPRM = µ-Opioid Receptor; CCR5 = Chemokine Receptor 5; GRM1 = Metabotropic Glutamate Receptor 1; PTH1R = Parathyroid Hormone-1 Receptor; DRD2 = Dopamine D_2_ Receptor; 5HT7R = Serotonin 5-HT_7_ Receptor; FFAR1 = G Protein-Coupled Receptor 40; H4 = Histamine H_4_ Receptor; SSR2 = Somatostatin Receptor Subtype-2; 5HT2B = Serotonin 5-HT_2B_ Receptor; GPR139 = G protein-Coupled Receptor 139; ADORA1 and ADORA2 = Adenosine A_1_ and Adenosine A_2A_ Receptors, respectively; GABABR1 = GABA_B_ Receptor.

## Data Availability

Data sharing not applicable.

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
