# Peer review of "Mind the Gap—Deciphering GPCR Pharmacology Using 3D Pharmacophores and Artificial Intelligence"

_pharmaceuticals, 2022, doi:10.3390/ph15111304_

Round 1

Reviewer 1 Report

The manuscript entitled “Mind the Gap – Deciphering GPCR Pharmacology Using 3D

Pharmacophores and Artificial Intelligence” is well designed and written.

It draws attention to the role of 3D pharmacophore modeling in the research on GPRC function and structure. In my opinion, all the most important issues have been raised.

 I recommend the publication of the manuscript in Pharmaceuticals after revision on the following issues:

-        In paragraph 4.2 (lines 565-597) the authors refer directly to the specific compounds described in the article by Ishida et al. This should be changed or the structures should be given in a separate figure.

-        I have also minor comments:

Please carefully check the references sections, especially highlighted positions in the attached pdf file. The references in the bibliography must be unified: in some items, the names of the authors are given, not the initials; the full name of the journal is given in some items, and most cases only an abbreviation; in a few items the name of the journal or other data was not given at all, and there was no milk in many items.

There are also some English changes required (also highlighted in the manuscript).

Reviewer 2 Report

The authors are presentning a detailed and easy to follow review on GPCR pharmacology using in-silico state of the art techniques.

The manuscript covers all possible issues on the subject. It is well written and could be a very nice handbook for young or  senior scientists.  Well written review, easy to follow the history of GPCRs and Virtual Screening campains.

I have some minor suggestions/correction to make:

1) on lines 109-114 rephrase big sentence such as:

 When an atomistic model of the target is avaiable, usually derived from X-ray crystallography, nuclear magnetic resonance (NMR) spectroscopy, cryo-electron microscopy (cryo-EM), homology modeling or machine learning prediction algorithms such as AlphaFold [16], structure-based 3D pharmacophores can be generated from apo binding sites or from structures of ligand-target complexes for more accurate drug design. 

2) 4.2 Compound 13 must not been noted with number as in prototype article. Use some description of the chemical skeleton instead.

 3) I would like some comparison between the different models for the same GPCR. For example 5HT2BR has been studied using Scaffold hopping and Hit-to-lead Optimization and Dynamics in GPCR-based pharmacophore modeling. The first one is more quick and the second is more accurate to GPCR flexibilty but needs more time to complete.

4)  On section 6 during the introduction the authors should mention that when using data from different laboratories there is need to normilize data. Thus on Chembl database they have introduced the p-chembl value (−log10 (molar IC50, XC50, EC50, AC50, Ki, Kd or Potency). However on Discussion the authors are raising this issue without more details.

5) Moreover i would suggest to emphysize that ML algorithms are mostrly utilized as filter on nover designed molecules and they can't predict new scaffolds. Thus using data from graph-theory can't go back and suggest new scaffold or substitutions.

6) lines 739. Use brackets for reference instead of parentheses.

Reviewer 3 Report

 Thanks Authors for submitting the review:  Mind the Gap – Deciphering GPCR Pharmacology Using 3D Pharmacophores and Artificial Intelligence.

I would suggest to consider within key words also: G protein-coupled receptors (GPCRs) 

This elegant review covers the field of G protein-coupled receptors (GPCRs) jet with significant gaps in understanding of their nuanced structure and function. Authors focus on 3D pharmacophore models,  powerful  computational tools in in silico drug discovery, presenting myriad opportunities for the integration of GPCR structural biology and cheminformatics. The review highlights possible  application of 3D pharmacophore modeling to de novo drug design, discovery of biased and allosteric ligands, scaffold hopping, QSAR analysis, hit-to-lead optimization, GPCR de-orphanization, mechanistic understanding of GPCR pharmacology and elucidation of ligand-receptor interactions. The  review  analyze challenges in the field of GPCR drug discovery, detailing how 3D pharmaco- phore modeling can be used to address them, presenting  opportunities afforded by 3D  pharmacophore modeling in the advancement of  understanding and targeting of GPCRs. 
